# Caffeine, Paraxanthine, Theophylline, and Theobromine Content in Human Milk

**DOI:** 10.3390/nu14112196

**Published:** 2022-05-25

**Authors:** Aleksandra Purkiewicz, Renata Pietrzak-Fiećko, Fritz Sörgel, Martina Kinzig

**Affiliations:** 1Department of Commodity Science and Food Analysis, Faculty of Food Sciences, University of Warmia and Mazury in Olsztyn, 1 Cieszyński Square, 10-726 Olsztyn, Poland; aleksandra.purkiewicz@uwm.edu.pl; 2Institute for Biomedical and Pharmaceutical Research, 90562 Nürnberg-Heroldsberg, Germany; ibmp@osn.de (F.S.); kinzig@ibmp.osn.de (M.K.); 3Institute of Pharmacology, Faculty of Medicine, University Duisburg-Essen, 45117 Essen, Germany

**Keywords:** methylxanthines, breast milk analysis, lactation, caffeine metabolites

## Abstract

This study aimed to assess the content of caffeine and its metabolites—paraxanthine, theophylline, and theobromine—in breast milk according to selected factors. Samples of human milk were collected from 100 women living in the east–northeast region of Poland. Information on the consumption of beverages and foods containing caffeine was collected using a 3 day food record. The determination of caffeine and its metabolite content was performed using liquid chromatography–mass spectrometry (LC–MS/MS). This study research showed that more caffeine was found in the milk of women living in cities, with secondary education, aged 34–43, and also in milk from the 3rd and 4th lactation periods (*p* ≤ 0.05). Factors such as place of residence, level of education, age, and stage of lactation influenced the nutritional choices of breastfeeding women, which had an impact on the level of caffeine and its metabolites in breast milk. A positive correlation was found between the consumption of caffeine with food and drinks and its level in human milk.

## 1. Introduction

Human milk is a natural and superior food for infants, requiring the optimal composition to meet their nutritional needs early on in life and providing associated immunological and psychological benefits. It is known that breastfeeding reduces child mortality and has health benefits that extend into adulthood. Many global bodies, including the American Academy of Pediatrics (AAP), the American College of Obstetricians and Gynecologists (ACOG), and the World Health Organization (WHO), recommend breastfeeding infants for at least the first 6 months of life and continuing breastfeeding with appropriate complementary foods to at least until 2 years of age [1,2].

Proper nutrition in early life is imperative for the physical and mental development of the child; moreover, bioactive compounds contained in breast milk are essential for the child’s development and protect the young against diseases. The composition of breast milk is determined by stage of lactation, the mother’s heart health and factors such as diet, race and geographical environment [3]. On the other hand, in addition to ingredients that are beneficial to infants, compounds that potentially have a negative impact on the baby’s health and development can pass into breast milk. A group of such compounds are methylxanthines, including caffeine and its metabolites—paraxanthine, theophylline, theobromine—which accumulate in breast milk when consumed in food by the nursing mother [4,5]. On consuming caffeine, it is quickly absorbed from the gastrointestinal tract; and after 30–120 min, it reaches maximum level in the blood. Caffeine is transported in blood to the tissues. It crosses the blood–brain barrier and passes through the placenta to the fetus, amniotic fluid, and breast milk [5]. Excessive consumption of caffeine by a nursing mother can negatively affect a breastfed infant. Too high a dose of caffeine in the child’s body causes muscle tremors and increased muscle tension. Excess coffee consumption by nursing women causes nausea and irritability among infants, as well as nervousness, sleep problems, and convulsions. Researchers report that these symptoms in infants significantly decreased after reducing the share of caffeinated foods in the daily diet. The half-life of caffeine in infants, especially breastfed, is much longer than in an adult, and therefore poses a greater risk to infants [6]. 

Caffeine is metabolized in the liver and approximately 2% of caffeine is excreted in the urine in unchanged form, while its decomposition produces paraxanthine, theobromine, and theophylline, and the final metabolite is 1methyl uric acid [5,7,8]. The metabolism of caffeine is shown in Figure 1. Paraxanthine constitutes approximately 72% of caffeine metabolites, and theophylline and theobromine constitute 10 and 5%, respectively [8,9]. Theobromine is a metabolite of caffeine; moreover, it is the main methylxanthine contained in cocoa [10]. The ratio of caffeine to theobromine in cocoa is 1:5 [11]. Therefore, the presence of theobromine in the daily diet is determined to a greater extent by cocoa consumption than by coffee or tea.

Caffeine is a chemical compound found in beverages: coffee, tea, and energy drinks; and in food: chocolate, cocoa, and sweets [13,14]. Energy drinks and instant coffee contain the most caffeine in an average serving. In terms of tea, leaf tea, both black and green, contains more caffeine. Levels of caffeine are over 3-fold higher in dark chocolate than in milk chocolate [15] (Table 1). It is recommended that both pregnant and lactating women limit caffeine consumption to 250–300 mg per day in order to avoid complications to the child’s health [16,17]. 

Nutrition specialists emphasize that food choices are the result of many factors—physical (geographical location, season, level of economic development, and city development level), social (social class, level of education, and religion), and psychological (nutritional needs, psychophysical state, and physiological state). Malczyk [18] reports that women are the highest consumers of caffeine. Moreover, significantly more caffeine is recorded in the diet of people aged over 30 and living in rural areas. Rehm et al. [19] report that the consumption of caffeine, including coffee and tea, is dependent on race/ethnicity, level of education, and economic status. More caffeine-rich beverages are consumed by older age groups with higher incomes. Most of the available literature on caffeine consumption by nursing women focuses on its effects on infant health, but only a few authors have investigated the content of methylxanthines in human milk to date. The novelty of the undertaken research is determined by the analysis of the content of caffeine and its metabolites according to selected factors. Although the composition of macronutrients such as protein or carbohydrate in human milk is physiologically conditioned, the composition of trace compounds in human milk including caffeine is an individual matter and may depend on many predisposing factors. Based on the available literature, it was assumed that the level of dietary caffeine intake by lactating women influences content of caffeine and its metabolites in breast milk; dietary caffeine intake in lactating mothers is influenced by demographic factors such as educational attainment and rural/urban living. The aim of this study was to perform a comprehensive assessment of the contents of caffeine and its metabolites in breast milk according to level of education, place of residence, age, and stage of lactation.

## 2. Materials and Methods

### 2.1. Chemicals

Chloroform, formic acid, methanol, ammonium formate, tetrahydrofuran and the internal standard solution of caffeine were obtained from Sigma Aldrich (St. Louis, MO, USA). 

### 2.2. Materials

The experimental material included human milk. Samples of human milk were collected from 100 women living in the east-northeast region of Poland. The nursing women participating in this study did not suffer from any diseases and did not take any medications during this study. All women had experienced a natural delivery. In each of the women, delivery proceeded without complications for the health of the mother and child. Milk samples were collected from women aged 18–43 years. The characteristics of the studied population are presented in Table 2. The breastfeeding population was divided into age categories (18–25 years old, 26–33 years old, and 34–44 years old); level of education—higher (at least bachelor studies) and secondary (general or technical secondary school); place of residence—city (>30,000 inhabitants) and village (<30,000 inhabitants); and stage of lactation (1st—colostrum: breast milk until a few days after birth; 2nd—foremilk: breast milk until two weeks after birth; 3rd and 4th: hindmilk—breast milk from 4 weeks of breastfeeding). Nursing women were asked to monitor food and drink consumption using a 3 day food record [20]. The nutritional interview was conducted by correspondence; it consisted in collecting information on 3 day consumption of food and drinks including household measures. The women in this study were instructed on how to keep a food record, and the method used to obtain intake data was as reproducible as possible. Food and drink intake data were discussed with the women and clarified if there were gaps or ambiguities in the record. On this basis, information was collected on the types and amounts of food products containing caffeine consumed by women, which are presented in Table 3. Among the sources of caffeine, women consumed coffee, tea, and dark and milk chocolate. Women did not indicate energy drinks as part of their diet. Based on previous studies, it has been shown that the level of caffeine in human milk is the highest 60 min after consuming a beverage or a food product containing caffeine [21]. Due to on-demand feeding, it would be difficult to schedule a certain amount of time to consume a caffeinated product and collect the milk within a strict time to maximize the level of caffeine. Therefore, during this research, women were asked to collect milk immediately after feeding. Caffeine is rapidly metabolized in the liver to metabolites that remain in the body long after consumption of caffeinated foods. Caffeine and its metabolites are constantly in the body, so regular coffee consumption contributes to the maintenance of levels of caffeine in milk [22,23,24]. The women collected milk by themselves into previously sterilized dark-glass bottles. The women collected milk samples immediately after feeding the baby. Each of the women used an electronic breast pump to collect milk samples. Women collected milk samples with volumes ranging from 50 to 100 mL. Milk samples were collected from the women every day in the evening, stored in special containers maintaining the temperature, and then frozen and stored at a temperature of −30 °C.

### 2.3. Sample Extraction

Human milk samples were thawed and vigorously mixed. To 21.5 mL of a human milk sample, 150 µL of internal standard solution (13C3-caffeine, 300 ng/mL) was added. After centrifugation (11.000 rpm, 10 min) 1 mL of the fat-free liquid was mixed with 200 µL ammonium format buffer (1 M, pH 2) and 4 mL chloroform, mixed for 10 min, and centrifuged (3000 rpm, 10 min). The organic solvent was evaporated and redissolved using 200 µL of mobile phase under a stream of N2 at 45 °C. A volume of 20 µL of each sample was injected into the LC–MS/MS.

### 2.4. Liquid Chromatography–Mass Spectrometry

The chromatographic system consisted of a binary LC pump (L-6200A Intelligent Pump) and an Autosampler (CTC Combi Pal) set to +4 °C. The separation was achieved on a Phenomenex Aqua C18 column (5 µm, 50 × 4.6 mm) using isocratic elution at ambient temperature with 0.1% formic acid in water (90%), methanol (5%) and tetrahydrofuran (5%). Detection was performed using a SCIEX API 3000 triple quadrupole mass spectrometer equipped with a turbo ion spray interface (SCIEX, Concord, ON, Canada). High-purity nitrogen gas was used as a nebulizer, curtain, auxiliary, and collision gas. The spectrometer was operated in the positive ion mode for the detection of caffeine (*m*/*z* 195 → 138), paraxanthine (*m*/*z* 181 → 124), theophylline (*m*/*z* 181 → 124), theobromine (*m*/*z* 181 → 138), and the internal standard (*m*/*z* 198 → 140). Data acquisition was performed using RAD (version 2.6, PE Sciex, Thornhill, ON, Canada) and data processing using MacQuan (version 1.6, Perkin Elmer, Toronto, ON, Canada, 1991–1998). The precision of the spiked quality control samples for all compounds was below 10%, with an accuracy between 85% and 115%.

### 2.5. Statistical Analysis

The values were expressed as the mean ± standard deviation (SD). Normal distribution was tested with the Shapiro–Wilk test and the homogeneity of variance with the Levene test. Statistical analysis showed that the distribution was not normal and the variance was not homogeneous; therefore, comparison of the collected quantitative data between the distinguished groups, by place of residence, level of education, age, and stage of lactation, was performed using the non-parametric Kruskal–Wallis test and Dunn’s post hoc test. Linear Pearson’s correlation coefficients were calculated to show the relationships between obtained data. The level of significance was assumed at α ≤ 0.05. Statistical analysis was performed using Statistica 13.1 (Statsoft Inc., Tulsa, OH, USA).

## 3. Results

### 3.1. The Content of Caffeine, Paraxanthine, Theobromine, and Theophylline in Breast Milk According to Selected Factors

Table 3 summarizes the consumption of beverages and food containing caffeine by lactating women, collected using a 3 day food record. The amount and type of consumed beverages and food containing caffeine varied according to the selected factors—place of residence, level of education, age, and stage of lactation. City dwellers consumed more coffee and a higher percentage of women prepared a larger amount of coffee. Moreover, a higher percentage of women consumed dark chocolate. Women with secondary education consumed more coffee, and women with higher education limited their consumption. Among respondents aged 18–25, coffee was consumed twice a day, while almost 70% of respondents aged 34–44 consumed coffee more than twice a day. A total of 40% of women in the 1st lactation period consumed coffee once a day, while 17% of women declared that they do not consume coffee. Most coffee was consumed by women in the 3rd–4th period of lactation—almost 50% of them consumed coffee more than twice a day. Table 4 presents the content of caffeine and its metabolites—paraxanthine, theophylline, and theobromine—in breast milk according to selected factors, place of residence, level of education, age, and stage of lactation.

The content of caffeine and its metabolites was identified in each of the human milk samples. The dominant metabolite of caffeine in the milk samples was theobromine, while the lowest levels were identified for theophylline (*p* < 0.05).

In terms of place of residence, significantly higher levels of caffeine and its metabolites were noticed among women living in cities. Levels of caffeine, paraxanthine, theophylline, and theobromine in the milk of city dwellers were, respectively, 2.5-, 3-, 1.5-, and 2.5-fold higher compared to women living in villages. 

In terms of level of education among lactating women, results varied according to chemical compound. Higher levels of caffeine and paraxanthine were found in the milk of breastfeeding women with secondary education (*p* < 0.05), while higher levels of theophylline and theobromine were recorded in the milk of women with higher education. The content of caffeine and its metabolites was also analyzed in terms of age. In terms of the age range, the content of a given chemical compound differed significantly. The milk of women aged 34–44 years contained the highest levels of caffeine and its metabolites; levels of caffeine were 2-fold higher than in women aged 18–25 and 26–33 years. Further, levels of paraxanthine, theophylline, and theobromine in the milk of women aged 34–44 years were, respectively, more than 1.5-, 2.5-, and 2-fold higher than in the milk of women aged 18–25; and more than 2.5-, 3- and 2-fold higher than in the milk of women aged 26–33 years.

In the tested samples, the content of individual methylxanthines was compared according to stage of lactation. In terms of caffeine content, the highest amounts were noticed in milk from the 3rd and 4th lactation periods. The biggest differences in caffeine content in milk when comparing stage of lactation were noted in the 1st lactation period compared to the 3rd and 4th lactation periods—the levels were 2.5-fold higher in the latter (*p* < 0.05). In the 3rd and 4th stage of lactation, the caffeine content in breast milk was 2.5 times higher than in the first stage of lactation (*p* < 0.05). In terms of paraxanthine, theophylline and theobromine, there were no differences in the content of these compounds in milk from the 2nd, 3rd, and 4th lactation periods.

### 3.2. The Relationship between Caffeine Consumption and Caffeine Content in Human Milk

Figure 2 shows the correlation between the consumption of caffeine in food and beverages and its level in human milk. A strong positive correlation (*r* = 0.72) was demonstrated between the effect of caffeine consumption on the content of this methylxanthine in human milk, with significance at *p* < 0.05.

### 3.3. Association between Obtained Data

Table 5 shows the correlation between the content of caffeine and individual metabolites—paraxanthine, theophylline, and theobromine in breast milk.

There was a very strong positive correlation between caffeine and paraxanthine content (*r* = 0.92), significant at *p* < 0.05. There was a directly proportional relationship between the content of caffeine and theobromine. The correlation relationships between caffeine and theophylline and caffeine and theobromine were strongly positive (*r* = 0.73 and 0.64), significant at *p* < 0.05.

Figure 3 shows the distribution of the content of individual caffeine metabolites versus caffeine content. In terms of paraxanthine, it was observed that points clustered about the straight line, suggesting a linear relationship. The correlation plots of theophylline and theobromine show more dispersed points, hence the lower correlation coefficients.

## 4. Discussion

The World Health Organization (WHO) states that lifestyle is the main factor in living a healthy life and ensuring a good quality of life [25,26]. Among the many factors influencing the composition of breast milk, time of day, stage of lactation, mother’s diet, and socioeconomic and demographic conditions can be mentioned [27,28]. Urbanization contributes to changes in the pattern of food consumption, mainly due to different lifestyles. Bo et al. [29] found that the economic status of nursing mothers contributed to their fluctuating fruit consumption. Studies by other authors have shown that the diets of nursing mothers living in cities were more varied [30]. Place of residence significantly influences consumption; especially of stimulant food products, including coffee. Norms for caffeine consumption by nursing women vary—the World Health Organization recommends that caffeine consumption should not exceed 300 mg. In turn, the European Food Safety Authority and the UK National Health Service (NHS) report that the habitual consumption of caffeine by nursing mothers should not exceed 200 mg [6]. In the studies conducted, higher levels of caffeine and its metabolites were identified in the milk of women living in urban areas compared to inhabitants of rural areas. According to the data collected using a 3 day food record, it was observed that almost 60% of urban residents consumed coffee more than 2 times a day, while over 50% of rural residents consumed coffee only once daily. The difference is in lifestyles. It is reported that city dwellers often prefer to drink beverages that contain caffeine, which is a direct result of their fast pace of life, where coffee acts as a means of treating excessive fatigue. Moreover, urban areas have several cafes, confectioneries, and restaurants with numerous attractive caffeinated drinks, which may attract several consumers [31,32]. High theobromine content may be caused by the consumption of chocolate in the diet of lactating women. The United States Department of Agriculture (USDA) report confirms that city dwellers consume more caffeine (along with coffee and tea) due to the faster pace of life and more hectic lifestyles [33]. 

Education is an important factor in promoting health awareness. it is reported that knowledge determines nutritional behavior; especially in chronically ill people as well as pregnant and lactating women [34]. Food choices depend on many factors, including level of education [35]. It is reported that people with a higher education make more informed food choices and are more likely to know current dietary recommendations [36]. In the studies conducted, the milk of women with higher education showed lower levels of caffeine and each of the metabolites, which may suggest that women with higher education are more aware of nutrition during feeding. According to the collected data, more than half of women with secondary education consumed coffee more than twice a day. In addition, almost 30% of them prepared coffee with more than two teaspoons, which may contribute to the higher content of caffeine and its metabolites in the milk of women with secondary education compared to women with higher education. Almost 50% of women with higher education consumed coffee once a day and 50% of breastfeeding women prepared coffee with less than 2 teaspoons. It can be assumed that women with higher education may be more aware of the potential risk to the baby of excessively high caffeine consumption during feeding. Lawrence [37] reports that poorer eating habits correlate with a lower socioeconomic status. Among the socioeconomic factors, level of education is particularly important as it is the strongest derivative to good health. It has been proven that more educated people are more likely to engage in a healthy lifestyle, and such behavior has also been proven among pregnant and lactating women too [38].

Due to possible health complications for the baby, caffeine consumption should be limited during pregnancy and lactation [15,39]. McCreedy et al. [6] report that the consumption of chocolate and coffee by nursing mothers was associated with the occurrence of increased infant colic, as well as skin problems, including atopic dermatitis. The administration of a dose of 10 mg/kg of caffeine to an infant may cause respiratory distress, hyperactivity, hyperglycemia, hypertension, tachycardia, or hematuria. Newborns metabolize caffeine more slowly as compared to adults, and regular caffeine intake may maintain a high level of caffeine in the child’s body [40]. Other studies indicate that the administration of caffeine to newborns may negatively affect the skeletal system and induces osteoblast apoptosis. This results in a delay in the structural reconstruction of the tibial metaphysis and causes swelling of the mitochondria in osteoblasts [41]. In addition, caffeine abuse by a nursing mother may be associated with future weight problems in the baby. According to the literature, excessive exposure to caffeine is associated with a higher risk of obesity in children in later years [42]. The surveyed nursing women declared a different amount of caffeinated drinks consumed during the day. Some of them followed the WHO recommendations and maintained a frequency of less than 2–3 servings a day, but a significant proportion of women included larger amounts of these products in their diet. It is reported that pregnant and lactating women pay more attention to their diets and most follow dietary recommendations. Surveys of pregnant women conducted by Jarosz et al. [43] found that women consumed an average of 91 mg of caffeine per day. Only approximately 2% of them consumed levels of caffeine above the WHO dose of 300 g. Moreover, the main source of caffeine in the diet of pregnant women was black tea, not coffee. Referring to data recorded using a 3 day food record, coffee consumption by lactating women varied according to selected factors. The highest coffee consumption (more than 2 cups per day) was among urban dwellers, with secondary education, aged 34–44 years. Caffeine intake among these women ranged from 200 to as much as 400 mg/day. However, in most cases, total caffeine intake did not exceed the WHO recommended intake (250–300 mg). Tyrala and Dodson [40] studied the level of caffeine in blood serum after consumption—it ranged between 2.39 and 4.05 µg/mL at 2 h. The level of caffeine in milk was 1.1 µg/mL after 30 min and 1.43 µg/mL after 60 min. The milk of the studied women contained significantly lower levels of caffeine, and the highest identified amount was approximately 1 µg/mL, which indicates that the studied women did not abuse caffeine-containing food products.

Pregnant and lactating women do not have to give up caffeine consumption completely—the Food and Drug Administration placed caffeine in group B, which means that the substance can be used during pregnancy and breastfeeding only when necessary. The consumption of agents in group B should be minimized and, in addition to caffeinated foods, they include medications [44]. On the other hand, the Australian Drug Evaluation Committee placed caffeine in category C—it can be used by pregnant and lactating women when the benefit of the mother exceeds the potential risk to the child [41].

Verster and Koenig [16] report that coffee consumption is proportional to age and emphasize that people aged over 30 consume significantly greater amounts of caffeinated beverages, especially coffee. The highest level of caffeine and its metabolites was recorded in the milk of women aged 34–44 (*p* ≤ 0.05). Almost 70% of women aged 34–44 consumed coffee more than twice a day; moreover, half of these women prepared coffee with more than 2 teaspoons. Furthermore, tea consumption was more popular in this age group, with over 50% of women tea it more than twice a day. Other studies indicate that people in this age category consume more caffeine than young adults (20–24 years old), and one of the reasons for this is the greater levels of stress and more responsibilities in people of this age [45]. Jarosz et al. [43] also indicate that caffeine consumption increases with the age of pregnant and lactating women. The milk of lactating women aged 34–44 years contained more than 2 fold higher levels of theobromine, which may be contributed to by the fact that the highest percentage of women in this age declared consuming chocolate during the day, and 41% of breastfeeding women at this age preferred dark chocolate, which contains more theobromine than milk chocolate [46].

The studies conducted showed significant differences in the content of caffeine and its metabolites according to stage of lactation. The smallest levels of tested compounds were recorded in milk from the first lactation period. Colostrum is the first food of a newborn baby and has a unique composition, including immunological factors, which is why it is called “immune milk” [47]. During this period, mothers pay special attention to their diet, wanting provide the best milk to their baby. A high percentage of breastfeeding women pay special attention to adherence to dietary recommendations, especially in terms of the proper consumption of vegetables and fruits, whole grains, and regular meals [48]. Reports in the literature confirm that women pay special attention to their diet directly after childbirth if they decide to breastfeed their child [49]. In subsequent stages of lactation, when transitional and mature milk are formed [50], the content of caffeine and its metabolites was significantly higher than in the 1st lactation period. This may be due to the higher consumption of caffeinated drinks as an anti-fatigue measure. During the analysis of 3 day food records, it was found that women in the 1st lactation period restricted coffee consumption most—43% of them consumed coffee once a day, and 17% of them completely excluded coffee.

In the undertaken studies, the milk sample was not collected under strictly defined time conditions, but at the moment when the child needed food. The level of caffeine in milk fluctuates over time, peaking approximately 2 h after consuming a caffeinated product. It is not recommended to drink caffeinated beverages, including coffee and tea, immediately before feeding the baby, because that is when the level of caffeine in milk is at its highest [21]. It has also been shown that after 120 min, the level of caffeine in breast milk decreases. These results were obtained after women consumed 150 mg of caffeine. It has been shown that a double dose of caffeine (300 mg) causes a 2-fold increase in the level of caffeine in blood serum and breast milk. In this study, the timing of intake of caffeinated foods and beverages was not monitored, and caffeine content was assessed on the basis of whole-day habitual intake, without taking into account the highest level of caffeine in milk. Therefore, levels of caffeine milk may have varied according to the time interval between consumption of caffeine-containing foods or beverages and breastfeeding. Bojarowicz and Przygoda [41] report that by supplying the mother’s body with 35–336 mg of caffeine, the infant consumes its mother’s milk at an amount of approximately 1.3–3.1 mg. According to the literature, irritability and sleep problems were observed in infants whose mothers consumed approximately 10 cups of coffee a day [21]. None of the women in this study consumed these amounts of coffee; however, according to data collected by using 3 day food records, several breastfeeding women consumed higher total levels of caffeine per day—from more than 300 mg/day to as much as 400 mg/day—than levels recommended by the WHO (up to 250–300 mg/day). In addition to caffeine, each woman’s milk also contained certain levels of its metabolites, paraxanthine, theophylline, and theobromine. In the examined samples of human milk, the highest levels of a metabolite were identified for theobromine, which ranged from 56% to 88% of all caffeine metabolites. Theobromine is a metabolite of caffeine, and it is also the main active compound in chocolate. Theobromine, in contrast to caffeine, has a weaker effect on the central nervous system—it only moderately raises alertness and also has a relaxing effect on muscles. Although paraxanthine is the main metabolite of caffeine, the highest levels of a metabolite in the milk of the studied women were determined for theobromine, due to chocolate consumption. It was reported that a low consumption of chocolate by nursing mothers did not adversely affect the infant, but including more cocoa in the diet affected the infant. It has been shown that a daily amount of 250 g of cocoa and chocolate in the diet of pregnant and lactating women caused irritability in the child, and the symptoms began to decrease after restricting the consumption of chocolate [51]. Paraxanthine, which is the main metabolite of caffeine, increases lipolysis; therefore coffee is considered a dietary component that indirectly reduces body fat [52]. In addition, theophylline is widely used in chronic obstructive pulmonary disease and asthma treatment due to its effect on the respiratory system [53]. Caffeine, when not consumed in excess, has a beneficial effect on the body—it increases mental activity, improves mood and concentration, and prevents diseases (e.g., type 2 diabetes and Alzheimer’s disease), but this substance contains several compounds with a pharmacological effect and should be limited by people suffering from cardiovascular and gastrointestinal diseases as well as by pregnant and lactating women [22,54]. In terms of caffeine, its peak levels in breast milk are recorded approximately 1 h after its consumption; but in terms of its metabolites, other relationships have been found. The highest levels of paraxanthine are recorded 5–10 h after consuming caffeine-containing products, and that of theophylline and theobromine within 10 to 15 h [21]. 

When analyzing caffeine content in milk, it should be emphasized that the metabolism, clearance, and pharmacokinetics of caffeine depend on many factors that may affect the final content of caffeine and its metabolites in breast milk. One of the key factors is a genetic variation that affects the body’s response to caffeine. Some pregnant women may be more sensitive to its effects—they metabolize caffeine more slowly. There is also a group of people who metabolize caffeine quickly, based on the higher content of enzymes responsible for the metabolism of caffeine [55]. This factor was not analyzed in the undertaken studies, and it may affect the final results of the studies. Most drugs and active substances do not pass into the milk of lactating women in amounts that are dangerous for the baby. On the other hand, it is not without reason that an upper maximum dose that should be consumed by pregnant women (250–300 day mg/day) has been established. Regular consumption of caffeine, especially coffee, may reduce the iron content in breast milk, which has a negative effect on the nutritional value of breast milk. Moreover, chronic consumption of beverages and foods containing caffeine may lead, especially in the neonatal period, too levels of caffeine and its metabolites that are too high in the plasma of infants [56]. Research by Sasaki et al. [56] shows that women who consume more than 300 mg of caffeine a day during pregnancy are more likely to give birth to children with a low body weight compared to women who consume safe doses of caffeine. Wesselink et al. [57] showed in their studies that excessive consumption of caffeine in the pre-conception period may cause fertility disorders in both women and men, while Hoeven et al. [58] indicate a higher risk of cardiovascular disease in women who abuse caffeine-containing products during pregnancy.

Lastly, even though the vast majority of women follow the WHO recommendations and consume safe levels of caffeine during pregnancy and breastfeeding, there is still a percentage of women who abuse this active substance. In spite of growing nutritional awareness and compliance with nutritional recommendations during lactation, some women still do not follow the recommendations. Stage of lactation, just like the time of pregnancy, has a significant impact on the development and shaping of the baby’s body. Breast milk is the most recommended food for babies due to its unique composition. According to research, more and more breastfeeding women pay attention to a balanced diet and restrict the consumption of products that may have a potentially negative effect on the baby. Scientific societies clearly state that during feeding, the consumption of products containing caffeine should be limited [6,59,60]. The most common reason for consuming caffeine is chronic fatigue and stress that accompany a nursing mother, especially in the first months of a child’s life. The starting point for encouraging young mothers to reduce caffeine consumption is awareness raising related to nutrition during pregnancy and lactation in terms of possible health complications related to caffeine abuse. Today, nutritional programs or coaching applications offer methods of self-management of nutrition and lifestyle, which greatly facilitate the rational planning of meals for young mothers [61,62].

This study has certain limitations. Women fed their babies on demand, so levels of caffeine were not monitored following specific consumption of caffeinated food or drink. Therefore, because caffeine consumption was intermittent, the determined correlation coefficient between caffeine consumption and caffeine levels in breast milk may have been disturbed. The time between caffeine consumption and breastfeeding may have varied for each mother, therefore the plateau is reached extremely rapidly with caffeine consumption. However, during this study, the greatest possible consideration was given to the comfort of the mother and child, and therefore no modifications were made to the manner and frequency of feeding infants. The level of caffeine may also be affected by medications taken, and although the surveyed women declared that they do not take any medications constantly, temporary painkillers may react with caffeine, intensifying its effect. The rate of caffeine metabolism depends, among other things, on the condition of the liver, which is influenced by the amount of medicines consumed. In this study, we only included women who do not take medicine on a regular basis, but did not monitor their use of other medicines or dietary supplements, which can also affect liver function. There are some limitations related to food intake monitored by a 3 day food record in this study. The women declared the amount of milk and dark chocolate in their diet, but did not provide information about the possible consumption of other chocolate products (cakes and cookies). Consumption of cocoa-rich products can significantly affect levels of caffeine and theobromine in milk. Despite verification of the women’s dietary records, data on consumption of such products may have been underreported. Another aspect that may be considered a limitation of this study is the sample size of only 100 breastfeeding women, which was caused by the difficulty in recruiting women willing to test or insufficient amounts of milk to collect samples.

## 5. Conclusions

Stage of lactation is associated with diet among nursing woman as well as the exclusion or restriction of selected substances. Scientific societies indicate that caffeine during pregnancy and lactation can be consumed without adversely affecting the baby at a limited amount of 250–300 mg per day. The studies conducted show that selected factors, such as place of residence, level of education, age, and stage of lactation, have an impact on the content of caffeine and its metabolites in breast milk. In this study, higher levels of caffeine were found in the milk of women living in cities, with secondary education and age (34–44 years old). Moreover, the highest levels of caffeine were found in mature milk in the 3rd and 4th lactation periods. In addition to the studied factors determining the formation of caffeine and its metabolites in breast milk, its content may also be influenced by genetic variation, body physiology and diet during pregnancy and lactation. The choices and nutritional awareness of breastfeeding women depend to some extent on the mothers’ level of education, place of residence, age, and stage of lactation. According to the current knowledge, a moderate caffeine content in the diet of nursing mothers should not endanger infants. However, it is recommended to limit the consumption of caffeinated beverages and foods to avoid possible adverse effects on the child.

## Figures and Tables

**Figure 1 nutrients-14-02196-f001:**
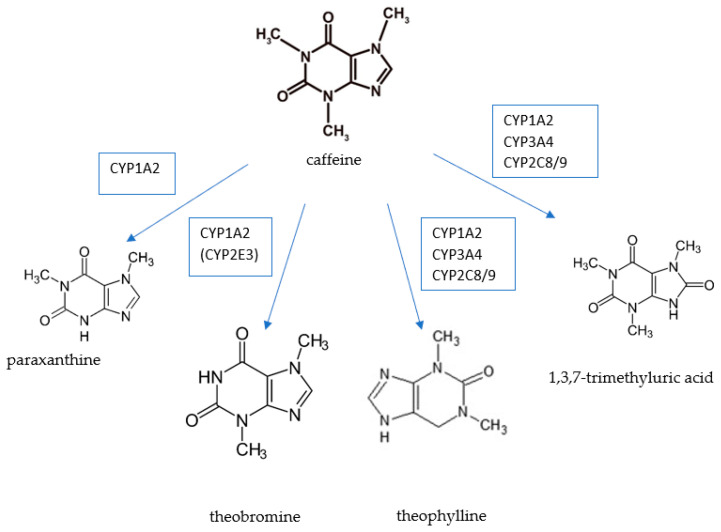
Caffeine metabolism. Source: own elaboration based on: [12].

**Figure 2 nutrients-14-02196-f002:**
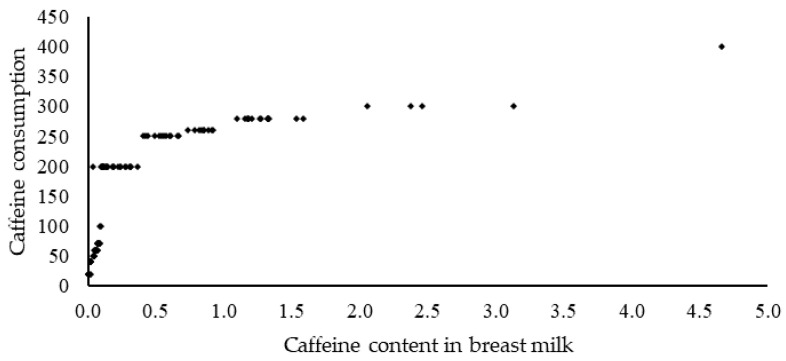
Correlation coefficients (*r*) calculated for the relationship between caffeine consumption and caffeine content inhuman milk (Correlation: *r* = 0.72).

**Figure 3 nutrients-14-02196-f003:**
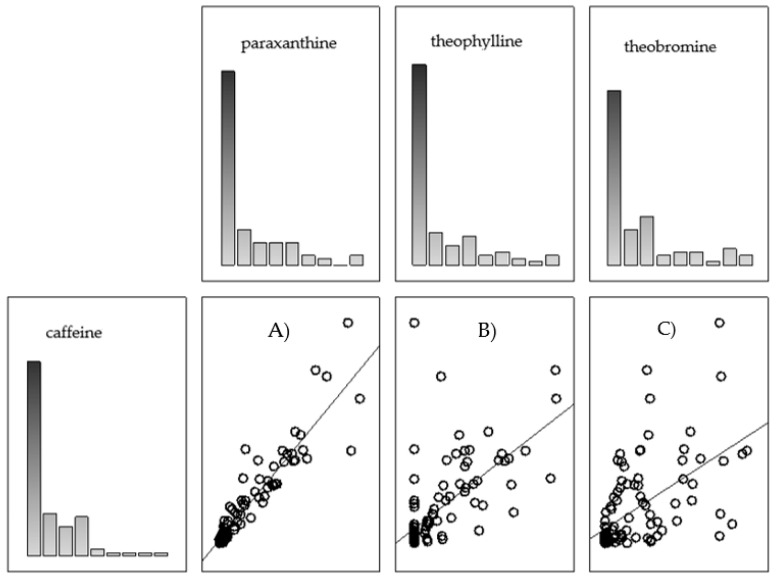
Graphical correlation coefficients (*r*) calculated for the relationship between caffeine content and the content of its metabolites—paraxanthine (**A**), theophylline (**B**), and theobromine (**C**).

**Table 1 nutrients-14-02196-t001:** Caffeine content in different beverages and food products. Source: own elaboration based on: [15].

Source of Beverage/Food	Volume/Weight	Caffeine Content [mg]
ground coffee	150 mL	60
instant coffee	150 mL	66
black tea	250 mL	
bags		31
leaf		43
green tea	250 mL	
bags		34
leaf		41
cacao	250 mL	5
Coca-Cola	250 mL	25
energy drinks	250 mL	80
bitter chocolate	100 g	67
milk chocolate	100 g	21

**Table 2 nutrients-14-02196-t002:** Characteristics of the studied breastfeeding women.

Factors	*n* = 100
Age, year	
18–25	29
26–33	37
34–44	34
Education	
Secondary	60
Higher	40
Place of residence	
City	65
Village	35
Stage of lactation	
1st	35
2nd	35
3rd and 4th	30

**Table 3 nutrients-14-02196-t003:** Information on the consumption of food products and beverages rich in caffeine collected using a 3 day food record.

	Place of Resident		Level of Education		Age			Stage of Lactation			*p*
Factor	City	Village	Secondary	Higher	18–25	26–33	34–44	1st	2nd	3rd–4th	
	***n* = 65** ***n*;%**	***n* = 35** ***n*;%**	***n* = 60** ***n*;%**	***n* = 40** ***n*;%**	***n* = 29** ***n*;%**	***n* = 37** ***n*;%**	***n* = 34** ***n*;%**	***n* = 35** ***n*;%**	***n* = 35** ***n*;%**	***n* = 30** ***n*;%**	
Coffee consumption *											
>than 2 times a day	38;58	6;17	33;55	13;33	9;31	8;22	23;68	3;9	10;29	14;47	
2 times a day	14;22	8;23	17;28	5;13	10;34	12;32	7;21	11;31	17;49	12;40	**0.0012**
1 time a day	8;12	19;54	6;10	19;48	8;28	12;32	4;12	15;43	7;20	4;13	
I don’t consume coffee	5;8	2;6	4;7	3;8	2;7	5;14	-	6;17	1;3	-	
Number of teaspoons (5 g) used for making coffee											
>than 2	29;45	9;26	17;28	6;15	9;31	8;22	17;50	2;6	9;26	11;37	
<than 2	21;32	13;37	26;43	19;48	12;41	14;38	12;35	15;43	17;49	14;47	0.5451
1 or less	10;15	11;31	13;22	12;30	6;21	10;27	5;15	17;49	8;23	5;17	
I don’t consume coffee	5;8	2;6	4;7	3;8	2;7	5;14	-	6;17	1;3	-	
Tea consumption **											
>than 2 times a day	27;42	14;40	25;42	14;35	7;24	9;24	19;56	9;26	11;31	9;30	
2 times a day	32;49	19;54	32;53	19;48	16;55	23;62	12;35	14;40	16;46	13;43	**0.0191**
1 time a day	6;9	2;6	3;5	7;18	6;21	5;14	3;9	12;34	8;23	8;27	
The type of tea consumed											
bags	57;88	32;91	55;92	31;78	25;86	28;76	29;85	31;89	27;77	24;80	**0.0291**
leaf	8;12	3;9	5;8	9;23	4;14	9;24	5;15	4;11	8;23	6;20	
Tea brewing time											
<than 10 min	62;95	31;89	57;95	36;90	27;93	32;86	31;91	35;100	32;91	26;87	**0.0002**
>than 10 min	3;5	4;11	3;5	4;10	2;7	5;14	3;9	-	3;9	4;13	
Chocolate consumption ***											
>than 3 cubes a day	-	-	-	-	-	-	-	-	-	-	
3 or less than 3 cubes a day	58;89	21;60	47;78	32;80	21;72	27;73	31;91	23;66	30;86	26;87	**0.0492**
I don’t consume chocolate	7;11	14;40	13;22	8;20	8;28	10;27	3;9	12;34	5;14	4;13	
The type of chocolate consumed											
bitter (70% cocoa and more)	23;35	4;11	7;12	15;38	12;41	12;32	14;41	13;37	8;23	11;37	
milk (less than 70% cocoa)	35;54	17;49	40;67	17;43	9;31	15;41	17;50	10;29	22;63	15;50	0.8192
I don’t consume chocolate	7;11	14;40	13;22	8;20	8;28	10;27	3;9	12;34	5;14	4;13	

* A portion of coffee—200 mL. ** A portion of tea—250 mL. *** 1 cube of chocolate = 5 g. *p* < 0.05 was indicated by using bold.

**Table 4 nutrients-14-02196-t004:** The content of caffeine, paraxanthine, theophylline, and theobromine in breast milk according to selected factors.

Factors		Compounds		
	Caffeine [µg/mL]	Paraxanthine [µg/mL]	Theophylline [µg/mL]	Theobromine [µg/mL]
Place of residence urban areas	0.590 ± 0.114 ^a^	0.355 ± 0.098 ^a^	0.028 ± 0.012 ^a^	0.442 ± 0.088 ^a^
rural areas	0.227 ± 0.012 ^b^	0.115 ± 0.055 ^b^	0.017 ± 0.010 ^b^	0.181 ± 0.062 ^b^
Level of education				
secondary	0.673 ± 0.032 ^a^	0.426 ± 0.021 ^a^	0.026 ± 0.009 ^a^	0.368 ± 0.012 ^b^
higher	0.545 ± 0.055 ^b^	0.331 ± 0.037 ^b^	0.029 ± 0.004 ^a^	0.442 ± 0.076 ^a^
Age (years) 18–25	0.485 ± 0.022 ^b^	0.303 ± 0.189 ^b^	0.019 ± 0.001 ^b^	0.278 ± 0.119 ^c^
26–33	0.353 ± 0.087 ^c^	0.190 ± 0.027 ^c^	0.016 ± 0.012 ^b^	0.313 ± 0.078 ^b^
34–44	0.838 ± 0.065 ^a^	0.505 ± 0.071 ^a^	0.048 ± 0.033 ^a^	0.630 ± 0.014 ^a^
Stage of lactation 1st	0.312 ± 0.167 ^c^	0.169 ± 0.057 ^b^	0.013 ± 0.009 ^b^	0.243 ± 0.082 ^b^
2nd	0.698 ± 0.095 ^b^	0.427 ± 0.032 ^a^	0.038 ± 0.011 ^a^	0.537 ± 0.021 ^a^
3rd–4th	0.794 ± 0.008 ^a^	0.472 ± 0.022 ^a^	0.0410 ± 0.055 ^a^	0.544 ± 0.059 ^a^

Explanation: means with different letters (^a,b,c^) in the columns are significantly different (*p* < 0.05).

**Table 5 nutrients-14-02196-t005:** Correlation coefficients (*r*) calculated for the relationships between the content of caffeine and its metabolites (*r* marked by * are statistically significant at *p* < 0.05).

	Paraxanthine	Theophylline	Theobromine
Caffeine	0.92 *	0.73 *	0.64 *

## Data Availability

The data presented in this study are available on reasonable request from the corresponding author.

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
