# Peer review of "Caffeine, Paraxanthine, Theophylline, and Theobromine Content in Human Milk"

_nutrients, 2022, doi:10.3390/nu14112196_

Round 1
Reviewer 1 Report
This is an interesting topic and the investigators have accumulated valuable data. However, the presentation of the results could be improved.
The most important information is missing from the results. We need to see the correlation between total caffeine intake and levels in human milk.
Introduction: It is currently rather long and could be made more precis by focusing on the hypothesis behind the study. I assume that this is:
"The level of dietary caffeine intake by lactating women influences breast milk content of caffeine and its metabolites with potentially adverse effects on the infant."
A secondary hypothesis could then be:
"Dietary caffeine intake in lactating mothers is influenced by demographics such as educational attainment, rural/urban living etc"
The section on metabolism of caffeine is long and detailed and would be better placed in the discussion. Figure 1 adds nothing to the paper. It would be better to provide and algorithm of the metabolism in a figure which will can illustrate a briefer text. A comment on the effects of maternal co-intake of medications and her liver health in particular should be considered.
Methods: There is a need for more detail about the mothers and the method of collection of samples. I assume the mothers were all healthy, not taking any medications and that the infants all had normal deliveries. This needs to be declared. Milk was collected immediately (I assume) after a breast feed and not after the mother had a drink. What method was used - breast milk extractor, manual expression etc? The discussion will need to elaborate on this as method of collection can have considerable effects on concentration of any constituents. Hind-milk can be very different from the fore-milk.
There are publications on standardisation of methods for human milk studies
Boix-Amoros A et al Nutrition Reviews 2019. doi 10.1093/nutrit/nuz019
Blyuss O et al Nutrients 2019. doi 10.3390/nu 11102416
With regard to the 3 day diet diaries was the volume of caffeine containing drink recorded and timing of intake in relation to the collection of milk?
Table 2 provides data on stages of lactation but does not define the stages. Was stage 1 colostrum? In addition there are significances all <0.01. This seems meaningless in relation to the data. What statistic on which data?
The MS standardisation section should clarify what is meant by precision. I assume that this means the <10% variability of levels assayed from samples spiked with known amounts (?how many)
Results: As indicated in the general comments there is a need to show the correlations between intake and milk content. This is the most important information. This can be followed by explaining the maternal demographics which are associated. This then gives practical information on which mothers should be targeted to provide advice on dietary control.
I was surprised that table 3 did not include data on energy drink intakes as they have the highest caffeine content. Furthermore they have become very popular particularly with younger people.
Statistical analysis must account for the number if tests
Discussion: I would like to see more detailed information on the adverse consequences of high caffeine in human milk on infant outcomes both short and long term. This after-all is the reason for the study. I assume the the investigators will be able to monitor this in the cohort they have studied.
Limitations of the study need to be discussed. Interactions between caffeine and its metabolites with other human milk constituents could modify effects and the health, medications and other environmental issues may equally cause interactions.
Author Response
Response to the comments from the reviewers
We would like to thank the reviewers for careful reading of the manuscript titled: The caffeine, paraxanthine, theophylline, and theobromine content in human milk.
All manuscript has been checked and revised by authors and the mistakes have been changed according to the reviewers comments. All corrections have been made (marked up using the “Track Changes” function) in the attached file of the manuscript.
Our response underneath describes point by point the changes made.
COMMENTS FROM THE REVIEWERS:
- This is an interesting topic and the investigators have accumulated valuable data. However, the presentation of the results could be improved.
The most important information is missing from the results. We need to see the correlation between total caffeine intake and levels in human milk.
Response: It was added a graph showing the correlation between consumption of caffeinated foods and beverages and caffeine levels in breast milk.
- Introduction: It is currently rather long and could be made more precis by focusing on the hypothesis behind the study. I assume that this is:
"The level of dietary caffeine intake by lactating women influences breast milk content of caffeine and its metabolites with potentially adverse effects on the infant."
A secondary hypothesis could then be:
"Dietary caffeine intake in lactating mothers is influenced by demographics such as educational attainment, rural/urban living etc"
Response: Reviewer comments were followed and the introduction was shortened. The detailed section on caffeine metabolism was shortened. In addition, research hypotheses were added as suggested.
- The section on metabolism of caffeine is long and detailed and would be better placed in the discussion. Figure 1 adds nothing to the paper. It would be better to provide and algorithm of the metabolism in a figure which will can illustrate a briefer text. A comment on the effects of maternal co-intake of medications and her liver health in particular should be considered.
Response: As suggested by the Reviewer, the section on caffeine metabolism was shortened. In addition, Fig 1 showing structural patterns of caffeine and its metabolites was removed from the manuscript. A schematic drawing of caffeine metabolism was placed in Fig 1. Information regarding the effects of drugs on liver function was added to the manuscript.
- Methods: There is a need for more detail about the mothers and the method of collection of samples. I assume the mothers were all healthy, not taking any medications and that the infants all had normal deliveries. This needs to be declared. Milk was collected immediately (I assume) after a breast feed and not after the mother had a drink. What method was used - breast milk extractor, manual expression etc? The discussion will need to elaborate on this as method of collection can have considerable effects on concentration of any constituents. Hind-milk can be very different from the fore-milk.
There are publications on standardisation of methods for human milk studies
Boix-Amoros A et al Nutrition Reviews 2019. doi 10.1093/nutrit/nuz019
Blyuss O et al Nutrients 2019. doi 10.3390/nu 11102416
Response: In the "Materials and Methods" section, information on nursing mothers was added. Information was added that the women were healthy and did not need to take medications and that the women delivered their babies naturally without complications. Information regarding the collection of milk samples was also added. Milk samples were collected immediately after each feeding, collected from the women's homes each evening, and stored in temperature-controlled containers and then in freezer conditions.
Women at different stages of lactation were enrolled in the study; therefore, colostrum, interim milk, and mature milk were examined. The study analyzed the nutritional choices related to the lactation period a woman is in. According to the results of the study, information was obtained that women who are in the first period of lactation are more attentive to restrictions on caffeine intake while breastfeeding.
- With regard to the 3 day diet diaries was the volume of caffeine containing drink recorded and timing of intake in relation to the collection of milk?
Response: Information on the volume of caffeinated beverages is provided below in Table 3.
- Table 2 provides data on stages of lactation but does not define the stages. Was stage 1 colostrum? In addition there are significances all <0.01. This seems meaningless in relation to the data. What statistic on which data?The MS standardisation section should clarify what is meant by precision. I assume that this means the <10% variability of levels assayed from samples spiked with known amounts (?how many)
Response: In the Material and Methods section, lactation periods are defined. Thank you for the valid comment regarding statistics. As the Reviewer noted, statistics are meaningless in relation to the data, so they were dropped.
- Results: As indicated in the general comments there is a need to show the correlations between intake and milk content. This is the most important information. This can be followed by explaining the maternal demographics which are associated. This then gives practical information on which mothers should be targeted to provide advice on dietary control.
Response: A graph showing the relationship between caffeine intake with food and beverages and caffeine levels in breast milk has been added to the Results section.
- I was surprised that table 3 did not include data on energy drink intakes as they have the highest caffeine content. Furthermore they have become very popular particularly with younger people.
Response: Table 3 does not include data on the consumption of energy drinks because the women surveyed reported not consuming them during the feeding period. This information has been added in the Materials and Methods section.
- Statistical analysis must account for the number if tests
Response: Statistical analysis used 5 statistical tests - Shapiro-Wilk, Levene'a. Kruskal Wallis and Dunn's) and linear Spearman's correlation.
- Discussion: I would like to see more detailed information on the adverse consequences of high caffeine in human milk on infant outcomes both short and long term. This after-all is the reason for the study. I assume the the investigators will be able to monitor this in the cohort they have studied.
Response: As recommended by the Reviewer, more information on the short- and long-term health effects of caffeine on infants was added to the Discussion section.
- Limitations of the study need to be discussed. Interactions between caffeine and its metabolites with other human milk constituents could modify effects and the health, medications and other environmental issues may equally cause interactions.
Response: In the Discussion section, limitations related to the study were added. These included failure to monitor how long after ingestion of caffeine-containing products mothers collected milk; lack of information regarding intake of emergency medications during the study (e.g., analgesics) that may impair liver function and caffeine metabolism; and limited sample size.
We would like to thank the reviewer for their contributions and valuable comments which make it possible to improve our manuscript.
Reviewer 2 Report
This study aimed to assess the content caffeine and its metabolites in breast milk based on the information of 100 women’s breast milk. The authors also evaluation of age, socioeconomic, as well as stage of lactation in the content caffeine in breast milk. There are some concerns:
- How the milk was collected should be described with more detail, including collection time (morning, noon, or evening) and collection volume. The study women were asked to collect foremilk, hindmilk or total milk?
- The time of breast milk collection is undoubtedly an important factor contributing to the bias of this study. The discussion on this (Lines.404-416, Pages.11) is not enough. How the caffeine concentration may be varied by the time of milk collection should be clarified.
- How the dietary data were collected by the 3-day food record should be described with more detail. Was there any quality control? The authors only presented the frequency of food intake in the Results. How about the amount?
- The author stated the recommended caffeine consumption (250-300mg per day) several times, but the level of caffeine consumption in the study population was not reported. If the authors lacked corresponding data, they should acknowledge this limitation in the Discussion.
- The authors did not analyze the association of food intake with breast milk composition (caffeine and its metabolites). What is the reason?
- In Table2, the information should be more specific, for example, the definition of secondary education level and the stage of lactation should be added in the Method or in the footnote of Table
- Did the authors perform statistical test among groups for data in Table3? If yes, the corresponding P values should be reported. If not, the conclusion about the choices and nutritional awareness of breastfeeding women (Lines.508-510, Pages.13) is baseless.
Author Response
Response to the comments from the reviewers
We would like to thank the reviewers for careful reading of the manuscript titled: The caffeine, paraxanthine, theophylline, and theobromine content in human milk.
All manuscript has been checked and revised by authors and the mistakes have been changed according to the reviewers’ comments. All corrections have been made (marked up using the “Track Changes” function) in the attached file of the manuscript.
Our response underneath describes point by point the changes made.
COMMENTS FROM THE REVIEWERS:
This study aimed to assess the content caffeine and its metabolites in breast milk based on the information of 100 women’s breast milk. The authors also evaluation of age, socioeconomic, as well as stage of lactation in the content caffeine in breast milk. There are some concerns:
- How the milk was collected should be described with more detail, including collection time (morning, noon, or evening) and collection volume. The study women were asked to collect foremilk, hindmilk or total milk?
Response: In the "Materials and Methods" section, information was added regarding the time of milk collection from the women (evening) and the volume of milk samples the women were to collect (50-100 ml).
- The time of breast milk collection is undoubtedly an important factor contributing to the bias of this study. The discussion on this (Lines.404-416, Pages.11) is not enough. How the caffeine concentration may be varied by the time of milk collection should be clarified.
Response: In the Discussion section, information was added regarding possible variables of caffeine content in milk samples due to the lack of monitoring of the time interval between consumption of caffeinated foods and beverages and infant feeding. After a previous review of the available literature, freezer storage was shown to have no effect on changes in caffeine concentrations in breast milk.
- How the dietary data were collected by the 3-day food record should be described with more detail. Was there any quality control? The authors only presented the frequency of food intake in the Results. How about the amount?
Response: In the "Materials and Methods" section, information was added regarding the interview collection 3-day food record method. Included is information that women were instructed on how to prepare food intake data; in addition, data were reviewed with the study women.
- The author stated the recommended caffeine consumption (250-300mg per day) several times, but the level of caffeine consumption in the study population was not reported. If the authors lacked corresponding data, they should acknowledge this limitation in the Discussion.
Response: In the "Discussion" section, information was added regarding the overall caffeine intake of the female subjects.
- The authors did not analyze the association of food intake with breast milk composition (caffeine and its metabolites). What is the reason?
Response: In the Results section, information and a graph were added showing the effect of caffeinated food and beverage intake on caffeine levels in breast milk.
- In Table2, the information should be more specific, for example, the definition of secondary education level and the stage of lactation should be added in the Method or in the footnote of Table
Response: In the "Material and Methods" section, information regarding the definitions of each grouping factor (level of education, place of residence, stage of lactation) was clarified.
- Did the authors perform statistical test among groups for data in Table3? If yes, the corresponding P values should be reported. If not, the conclusion about the choices and nutritional awareness of breastfeeding women (Lines.508-510, Pages.13) is baseless.
Response: As suggested by the Reviewer, the authors performed a statistical analysis of the data in Table 3 to show statistical differences in caffeinated food and beverage consumption.
We would like to thank the reviewer for their contributions and valuable comments which make it possible to improve our manuscript.